Attributions of meteorological and emission factors to the 2015 winter severe
haze pollution episodes in China Jing-Jin-Ji area
Tingting Liu[1,2], Sunling Gong[2,4], Jianjun He[2], Meng Yu[3], Qifeng Wang[4], Huairui Li[4], Jie Zhang[3], Lei
Li[3], Xuguan Wang[3], Shuli Li[3], Yanli Lu[3], Haitao Du[5], Yaqiang Wang[2], Chunhong Zhou[2], Hongli Liu[2],
Qichao Zhao[4]
[1] School of Mechanical Engineering, Hangzhou Dianzi University, Hangzhou, China
[2] State Key Laboratory of Severe Weather & Key Laboratory of Atmospheric Chemistry of CMA,
Chinese Academy of Meteorological Sciences, Beijing, China
[3] Langfang Bureau of Environmental Protection, Langfang, Hebei, China
[4] Langfang Academy of Eco Industrialization for Wisdom Environment, Langfang, Hebei, China
[5] Langfang Bureau of Meteorology, Langfang, Hebei, China

## Abstract

Northern China in the 2015 winter month of December has witnessed the most severe air
pollution phenomena since the 2013 winter haze events occurred, which triggered the first ever
Red Alert in the air pollution control history of Beijing, with an instantaneous $PM_{2.5}$ concentration
over 1 mg m$^{-3}$. Air quality observations reveal that there exist large temporal-spatial variations of
$PM_{2.5}$ concentrations over Beijing-Tianjin-Hebei (Jing-Jin-Ji) area between 2014 and 2015.
Compared to 2014, the $PM_{2.5}$ concentrations over the area in 2015 decreased significantly except
in November and December, with an increase of 36% in December. Analysis shows that the $PM_{2.5}$
concentrations are significantly correlated with the local meteorological parameters in the Jing-
Jin-Ji area, such as the stable conditions, relative humidity, wind field etc.. A comparison of two
month simulations (December 2014 and 2015) with the same emission data was performed to
explore and quantify the meteorological impact on the $PM_{2.5}$ over the Jing-Jin-Ji area.
Observation and modeling results show that the worsening meteorological conditions are the
main reasons behind this unusual increase of air pollutant concentrations and the emission

1    control measures taken during this period of time have contributed to mitigate the air pollution

2    (~9%) in the region. This work provides a scientific insight of the emission control measures vs.

3    meteorology impacts for the period.

# 1. Introduction

Severe air pollution has been observed in China for the last 15-20 years, with an annual mean concentration of fine particular matter ($PM_{2.5}$) ranging from 80 to 120 $\mu g$ $m^{-3}$ and over 1000 $\mu g$ $m^{-3}$ during some heavy haze episode. Haze phenomenon has become a major pollution problem in many China cities (Han et al., 2013; Wang et al., 2015), which causes wide public concern and has an adverse impact on human health and environment (Gurjar et al., 2010; Kan et al., 2012). Therefore, it is necessary to comprehensively investigate the emission sources, meteorological factors, and other characteristics of the $PM_{2.5}$ pollution in China and provide more effective control measures (Wang et al., 2008; Zhang et al., 2014).

Since the strict control measures of air pollutants over the country were enforced in 2013 by the government, a steady decrease of air pollutant concentrations has been observed with an annual mean $PM_{2.5}$ concentration dropping from about 85 $\mu g$ $m^{-3}$ in 2014 to 80 $\mu g$ $m^{-3}$ in 2015 for Beijing, from 86 to 70 for Tianjin, and from 118 to 88 for Shijiazhuang (Three typical cities in North China, http://www.mep.gov.cn/gkml/). Meteorological conditions, especially the large-scale circulations, are important factors to determine the variations of pollution level (He et al., 2016a; Jia et al., 2015). Significant regional transport, caused by special meteorological conditions, is critical for the formation of 2015 winter severe haze in Beijing (Sun et al., 2016). A number of papers have tried to analyze the meteorological contributions (He et al., 2017; Liao et al., 2015; Wang et al., 2013; Zeng et al., 2014) for individual cases but could not consider the emission changes for a comprehensive analysis. The most recent consensus is that these decreases partially can attribute to the difference in the meteorological conditions but largely should be

attributed to the control measures taken. However, due to the complex interactions between
pollution sources and meteorology, the quantitative contributions for each factor remain to be
separated.
The year of 2015 was an unusual year in terms of air pollution situation in Northern China,
which was in the middle of an El Niño-Southern Oscillation (ENSO) event around the globe
(Varotsos et al., 2016). Unusual climate and extreme weather happened everywhere. It was
found that the El Niño event had a significant effect on air pollution in eastern China (Chang et
al., 2016). In the first half of 2015, a steady decrease in major air pollutants was observed
compared to those in 2014 in northern China. However, in the last two months, a dramatic
increase was observed. The $PM_{2.5}$ concentration reached as high as 1000 $\mu g\ m^{-3}$ in Beijing and
triggered the first ever Red Alert of severer air pollution in the city. Would this unusual increase
of air pollution have anything to do with special meteorological conditions and the El Niño event,
and what was the role of emission control measures being played in this?
This paper presents an analysis and modeling study of air pollution conditions in December
2015 in Beijing-Tianjin-Hebei (Jing-Jin-Ji) area located in North China and explores the major
reasons behind these unusual increases from both the meteorological and emission points of
views. To evaluate the contribution of meteorological factors toward the severe pollution in
December 2015, the wind speed convergence lines (WSCL), static wind frequency data and other
parameters in December 2015 were specifically investigated and compared with data for the
same period of 2014. An analysis of this heavy haze pollution episode was also simulated with
the Chinese Unified Atmospheric Chemistry Environment (CUACE) model (Gong and Zhang, 2008).
The aim of this study was to provide information on the impact degree and mechanism of
meteorology variations and emission changes on the PM$_{2.5}$ haze pollution in this region.

## 3 2. Methodology

The research starts with the analysis of air pollution levels between 2014 and 2015, with a
focus on the last month of each year. The difference lays the foundation for the investigation,
where the meteorology factors that mostly influence the air pollution levels such as the stable
conditions, wind speed and directions as well as the relative humidity are probed, which will give
a qualitative description of the reasons for pollution changes from 2014 to 2015. Based on EAR-
Interim reanalysis data from European Centre for Medium-Range Weather Forecasts (ECMWF),
the potential effect of ENSO on atmospheric circulation and air quality in Jing-Jin-Ji area is also
investigated robustly. In order to quantify the meteorology impacts, a modeling study is carried
with the same emission rates in the model for 2014 and 2015 where the pollution level changes
are considered to be caused by meteorology only. The impact of emission changes on air
pollution can then be inferred from the difference between the observed pollution level changes
and the modelled level changes only due to the meteorology.

## 16 3. Air Quality Observations

The observational pollution data used in this study were from the near real time (NRT)
monitoring stations of the Ministry of Environmental Protection across the Northern China
(http://www.cnemc.cn/), with hourly concentrations of six major pollutants: particulate matter
with an aerodynamic diameter of less than 2.5 μm and 10 μm (PM$_{2.5}$ and PM$_{10}$), sulphur dioxide
(SO$_2$), nitrogen dioxide (NO$_2$), carbon monoxide (CO) and ozone (O$_3$). The monthly and annual
mean concentrations of $PM_{2.5}$ in three typical cities (Beijing, Tianjin, and Shijiazhuang) and Jing-
Jin-Ji area were investigated. $PM_{2.5}$ concentration in Jing-Jin-Ji area represents the regional
average for 13 cities, i.e., Beijing, Tianjin, Shijiazhuang, Handan, Xingtai, Hengshui, Cangzhou,
Baoding, Langfang, Tangshan, Qinhuangdao, Chengde, and Zhangjiakou. Spatial distribution of
13 cities has been shown in Figure 1. Based upon the entire year data for 2014 and 2015, the
annual mean concentrations of $PM_{2.5}$ are overall in a decreasing trend. For three typical cities of
Beijing, Tianjin and Shijiazhuang, the annual mean $PM_{2.5}$ concentrations in 2015 are 5.7%, 18.5%
and 29.2% lower than those in 2014, respectively.

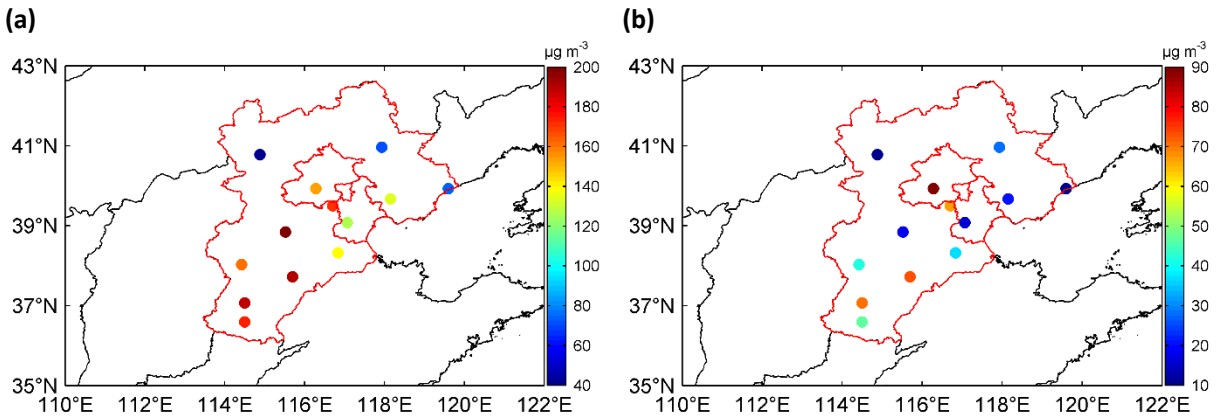

Figure 1:  (a) Monthly mean $PM_{2.5}$ concentrations in December 2015 and (b) and the change of
monthly mean $PM_{2.5}$ concentration in December between 2015 and 2014 over Jing-Jin-Ji area.
The regional mean $PM_{2.5}$ concentration over Jing-Jin-Ji area decreases 17.8%. The two year
monthly mean $PM_{2.5}$ concentrations (Fig. 2) indicate that from January to October, the
concentrations in 2015 are much lower than those in the same monthes in 2014. The unusual
increases of $PM_{2.5}$ concentration are found in the last two months, especially for December,
which is the focus of this study.
Regionally, the monthly mean PM$_{2.5}$ concentrations in December 2015 saw a large increase
compared to the same month in 2014, ranging from 5% to 137% in Jing-Jin-Ji area, with a mean
increase of 36% (Fig. 1). Beijing had the largest increase of 137%, jumping from approximately 61
$\mu$g m$^{-3}$ in 2014 to 145 $\mu$g m$^{-3}$ in 2015, while Qinhuangdao had the smallest increase of 5%, jumping
from approximately 69 $\mu$g m$^{-3}$ in 2014 to 72 $\mu$g m$^{-3}$ in 2015.

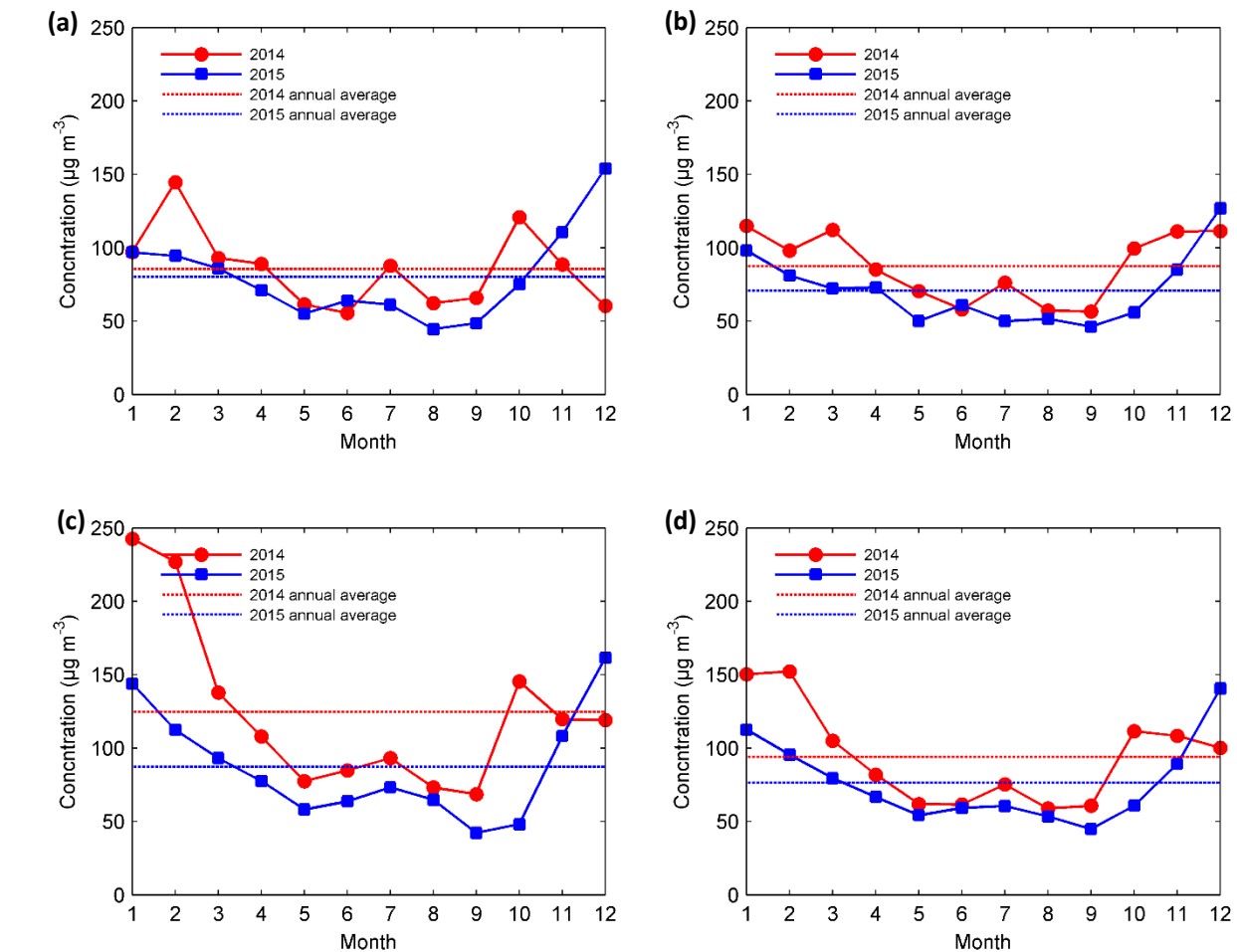

Figure 2: Comparison of monthly average PM$_{2.5}$ concentrations of 2015 and 2014 in (a)Beijing,
(b)Tianjin, (c) Shijiazhuang and (d)Jing-Jin-Ji area.

Certain factors must have had a dramatic change to cause this to happen. In view of the
steady decreases of air pollutants over Jing-Jin-Ji area in the first ten months of 2015, it can be
inferred that the emission reduction measures implemented in the region, including traffic
restriction and eliminating vehicles that fail to meet the European No. 1 standard for exhaust
emission, reducing coal consumption, forbidding straw burning, and reducing volatile organic
compounds (VOC) emission (http://bj.people.com.cn/n/2015/0526/c233088-25012933.html),
were effective in bringing the averaged concentrations of major pollutants down. In next session,
the meteorological conditions for the last month of 2014 and 2015 will be analyzed in details to
elucidate the reasons for this dramatic increase in Jing-Jin-Ji area.
## 4. Meteorology Factor Analysis
Closely related to the air pollution variations, meteorological conditions are the important
factors determining day-to-day variations of pollutant concentrations (He et al., 2016a). The
correlation between daily average $PM_{2.5}$ concentrations and four meteorological parameters, i.e.
2-m temperature (T2), 2-m relative humidity (RH2), 10-m wind speed (WS10) and boundary layer
height (BLH), is shown in Figure 3. The data processing in correlation calculation is the same as in
He et al. (2017).  $PM_{2.5}$ concentrations are positively correlated with T2 and RH2, while negatively
correlated with WS10 and BLH. The correlation coefficients are significant except correlation for
T2 in Shijiazhuang. The positive correlation between $PM_{2.5}$ concentrations and RH2 reveals the
importance of hygroscopic growth for PM in Jing-Jin-Ji area. The increase of WS10 and BLH
enhances the ventilation and diffusion capacity, and brings good air quality. The comparison of
correlation coefficients in three cities reveals that local meteorological condition has more
significant effect in Beijing than that in Tianjin and Shijiazhuang. Located in the northern edge of
the North China Plain, regional transport of pollutant in Beijing is less complex than other cities,
which may explain high correlation between PM$_{2.5}$ concentration and meteorological parameters
in Beijing.

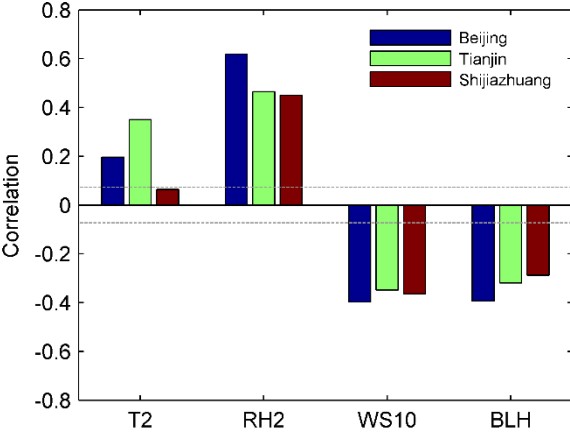

Figure 3: The correlation between daily average PM2.5 concentrations and daily average
meteorological parameters during 2014-2015. The dashed lines represent the critical correlation
coefficient that passes the t-test at a 95% confidence level.
Previous studies have shown that a major factor controlling the pollutant accumulation is
the atmospheric stability in association with the convergence at lower levels, which leads to the
accumulation of the polluted air from the surrounding areas and prevents pollutants from
diffusing away from the source regions (Liao et al., 2015; Wang et al., 2013; Zeng et al., 2014).
Therefore, the location of the convergence zone is critical in identifying the meteorological
conditions that are favorable or not for the formation of heavy pollution.
Two weather analysis maps are constructed based on average surface meteorological data
for December of 2014 and 2015 from China Meteorological Administration (CMA) (Figure 4). The
mean wind speed for December of 2014 reveals that large wind exists in Hebei province while
small wind speed exists in the south of Hebei province. Wind speed sheer, i.e., abrupt decrease
(increase) of wind speed, forms a convergence (divergence) zone, ie. WSCL. The WSCL locates in
the boundary of Hebei and Shandong provinces. A serious pollution band nearby the WSCL will
usually be formed due to the adverse dispersion conditions and pollutant accumulation.
Compared to 2014, the WSCL shifts to Beijing municipality and the center of Hebei province in
2015. The relocation of the WSCL results in the moving of pollution band northward and more
serious air pollution in Beijing and surrounding cities.

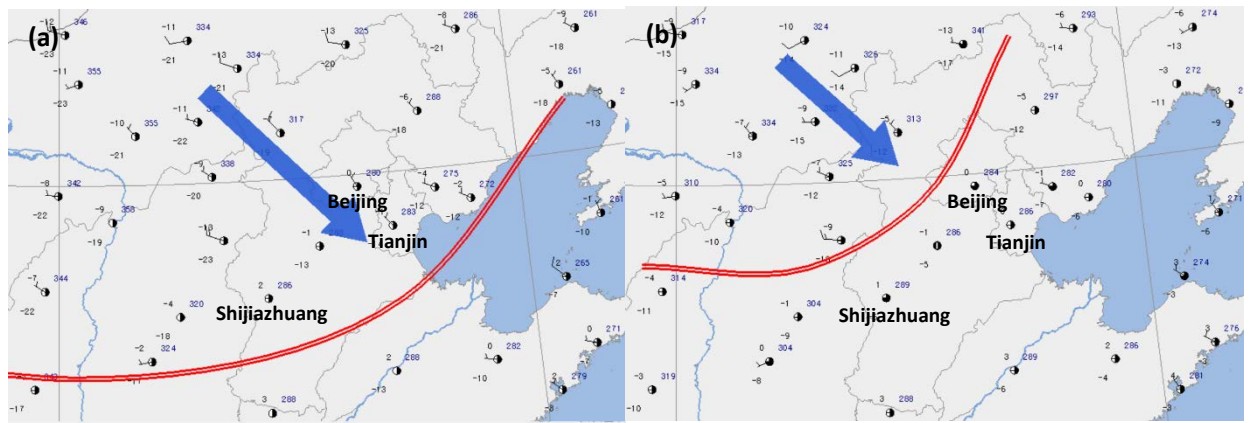

Figure 4: The weather analysis maps in December 2014 (a) and 2015 (b). Red line represents
WSCL.
14        Observational evidence has shown a teleconnection between the central Pacific and East Asia

during the extreme phases of ENSO cycles. This Pacific–East Asian teleconnection is confined to
the lower troposphere. The key system that bridges the warm (cold) events in the eastern Pacific
and the weak (strong) East Asian winter monsoons (EAWM) is an anomalous lower-tropospheric
anticyclone (cyclone) located in the western North Pacific (Wang et al., 2000). Si et al. (2016) has
found that during the 2015 El Niño period, the EAWM was weaker than normal during the 2015
winter with a temperature increase of 1.1 °C. The subtropical high was stronger and had a large
area than normal years (Li et al., 2016). As a consequence of the weaker EAWM, the cold front in
2015 could not extend to farther south than in 2014, leading to a northward shifting of the WSCL.

4       Chang et al. (2016) finds a close relationship between ENSO and air pollution in North China

in 2015. To deeply investigate the relation between ENSO and the air quality in North China, EAR-
Interim reanalysis data in December 1979-2015, including sea surface temperature (SST), mean
sea level pressure (MSL), 2-m temperature (T2), 10-m U and V wind component (U10 and V10),
were used. Area averaged SST anomalies (SSTA) over the Nino3 region (5°N-5°S, 150°-90°W)
provide an index typically used to represent ENSO variability (Tang et al., 2016). Time series of
monthly averaged SSTA over the Nino3 region are shown in Figure S3. Significant ENSO events
were found in 1982, 1997 and 2015. The MSL and 10-m wind anomalies over North China region
are shown in Figure 5. It seems that ENSO (SSTA>0) results in weak cold air and northerly wind,
while opposite for La Nina (SSTA<0). These relations indicate that the worse air quality in
December 2015 over North China may be related to the significant ENSO. Further study is needed
to investigate this correlation with longer period of data.

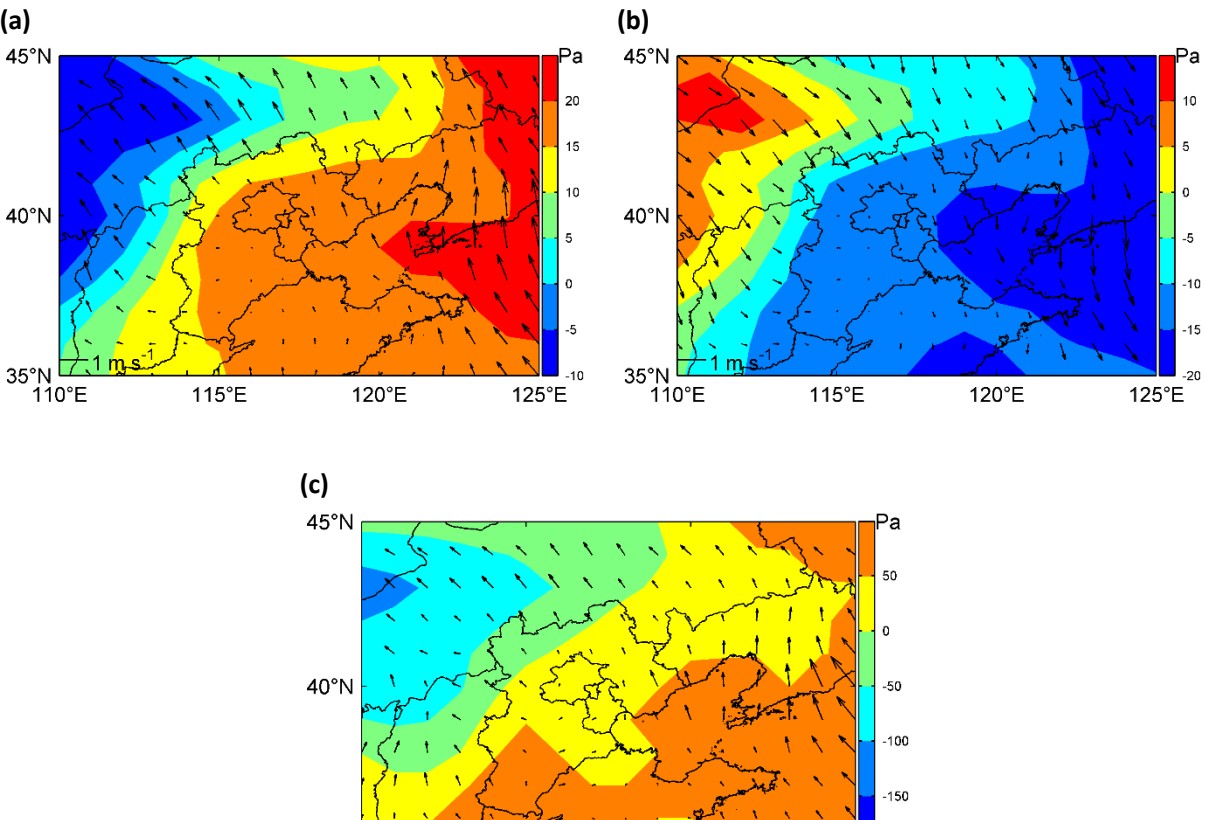

Figure 5: The MSL and 10-m wind anomalies over North China region. (a): SSTA larger than zero; (b) SSTA less than zero; (c) December 2015.

There are three consequences of the WSCL shifting. First of all, accompanied with the northerly shift of the WSCL is the shifting of the stable atmosphere zone to the central Hebei and Beijing areas in 2015, allowing the pollutants to easily accumulate along the lines. The observed static wind frequency (SWF, wind speed less than 1 m s$^{-1}$) distribution clearly supports this observation. Figure 6a is the regional distribution of SWF in December 2015, showing a high frequency along the convergence line, with the SWF changes from 2014 (Fig. 6b). Table 1 lists the SWF for three typical cities and regional mean over Jing-Jin-Ji area. Except for Shijiazhuang which had an unusual high SWF in 2014 and a decreasing SWF in 2015, other cities experienced an

increasing trend for stable weather. Impacted heavily by the shifting, Beijing and Tianjin had a
16-19% increase of SWF compared to 2014. Even with a decreasing trend for SWF, Shijiazhuang
had a similar SWF with other cities with more than half of the days (>50%) under static stable
environment.

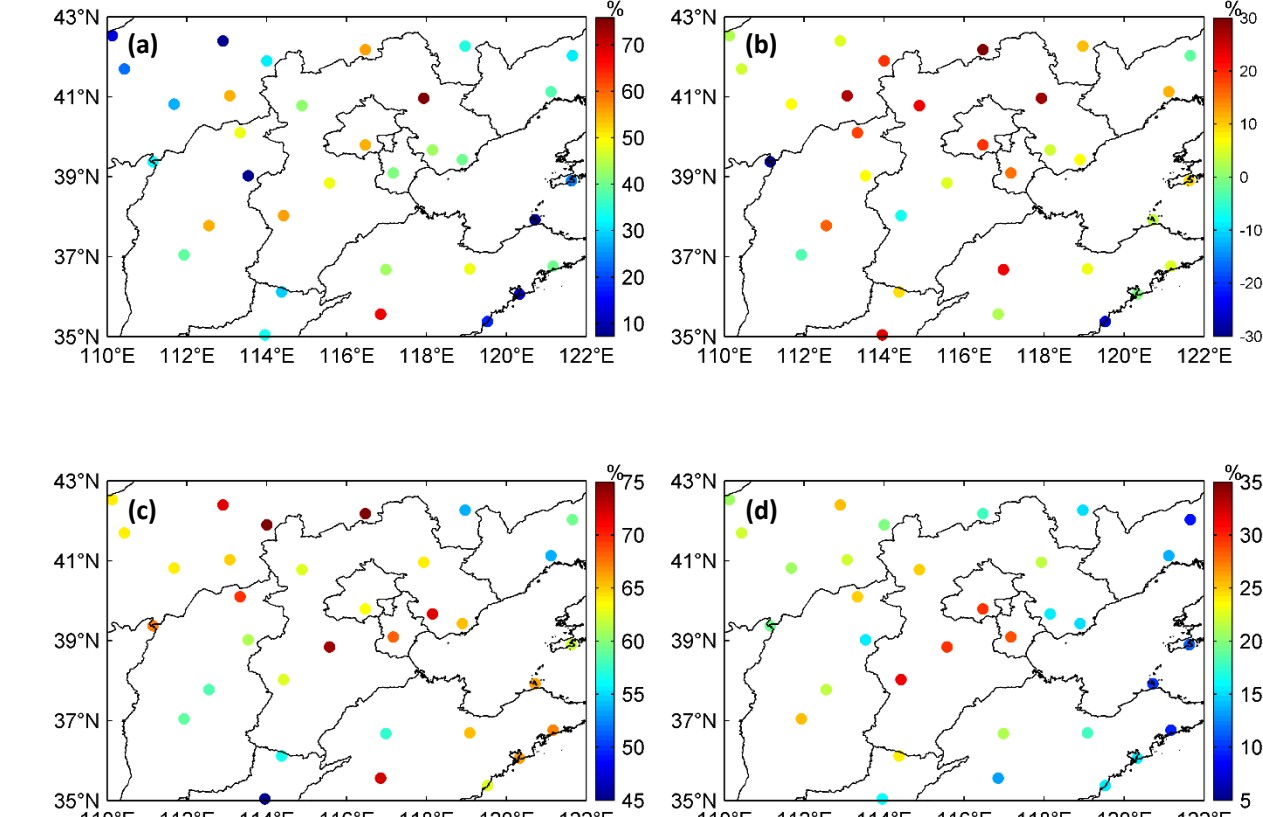

Figure 6: The observed SWF and RH2 distributions averaged for December in 2015 (a, c); Changes
from 2014 (b, d).
Table 1: Comparison of SWF (%), WS10 (m s$^{-1}$) and RH2 (%) for December 2015 and 2014

| City | Beijing | | | Tianjin | | | Shijiazhuang | | | Jing-Jin-Ji | | |
|------|------|------|-----|------|------|-----|------|------|-----|------|------|-----|
| | SWF | WS10 | RH2 | SWF | WS10 | RH2 | SWF | WS10 | RH2 | SWF | WS10 | RH2 |
| 2014 | 35 | 1.5 | 34 | 25 | 1.1 | 40 | 63 | 0.7 | 31 | 38 | 1.4 | 42 |
| 2015 | 54 | 0.5 | 64 | 41 | 0.6 | 68 | 55 | 0.6 | 63 | 50 | 0.8 | 67 |

In Beijing, the WSCL shifting in December 2015 not only increased the SWF but also changed
the wind directions. It is shown from Figure 7 that the north-west winds that usually diffuse the
air pollution away from Beijing were reduced by about 20% in December 2015 compared to the
same period in 2014, while the southerly wind frequencies were increased by 8% that brought
air pollution to Beijing. Compared to Beijing, the city of Shijiazhuang was not seen such a large
change (Fig. 7). The SWF in Shijiazhuang was reduced, and the northerly wind frequency even
increased by 6% in December 2015.

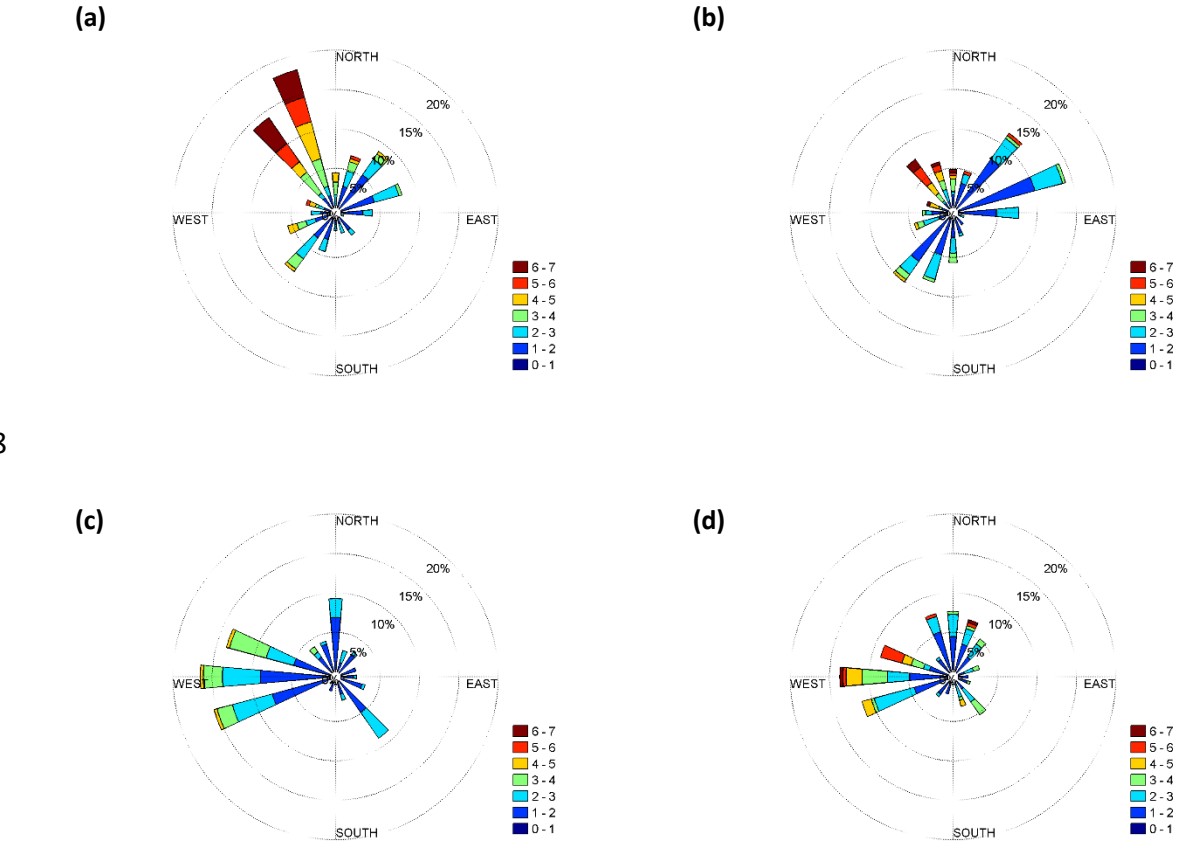

Figure 7: (a) The observed wind frequency and directions averaged for December 2014 and 2015
in Beijing (a-b) and Shijiazhuang (c, d) respectively.

1        The second consequence of the WSCL shifting is the northerly movement of moisture

from the South. Figure 6c and 6d shows the averaged RH2 for December of 2015 and changes
from 2014. It is obvious that as the shift of the WSCL to the North, the RH increases are primarily
on the north side of the WSCL with an increase of 30% in Beijing (other cities in Table 1). $PM_{2.5}$
concentration is positively correlated with RH2 (Figure 3). The impact of increasing RH has an
adverse influence on the visibility under the same loading of particulate matters and also
promotes the formation of secondary formation of particulate matters from gaseous species.
Because of the WSCL shifting, the increase of RH in Shijiazhuang was even slightly larger than that
in Beijing, at about 32%. Researches (Chang et al., 2009) have shown that the extent of $SO_2$
oxidation to sulfate and $NO_2$ oxidation to nitrate increased with the increase of relative humidity
during both of the episode daytime and nighttime pollution in Taiwan. Gund et al. (1991) found
that the oxidation rate of $SO_2$ to sulfate could increase by about 10 times if the relative humidity
increased from 40 to 80% in sea-salt aerosols. If $NO_2$ ($SO_2$:$NO_2$ = 1:1) was added to the gas phase,
the rate for example at a relative humidity of 40% - could be increased by 24 times, indicating
the enhanced conversion tendency of $SO_2$ to $PM_{2.5}$ by both relative humidity and $NO_2$ (Gund et al.
1991). Though the detailed mechanism of this enhanced oxidation in Northern China needs
further study, the increased relative humidity may partially be attributed to the decreases of $SO_2$
(from 86 $\mu$g m$^{-3}$ to 61 $\mu$g m$^{-3}$ in Jing-Jin-Ji area) during the heavy pollution months in 2015 winter
as compared to the same period of 2014.
The third consequence of the WSCL shifting is the decrease of BLH. Previous research (Liu et
al., 2010) has revealed that the invasion of cold air increases the turbulence flux and the BLH over
Beijing area. Compared to 2014, the weaker cold air in December 2015 resulted in the decrease
of BLH in the range from 50 to 300 m over Jing-Jin-Ji area (Figure 8), which was one of the main
reasons for heavy haze pollution in December 2015.

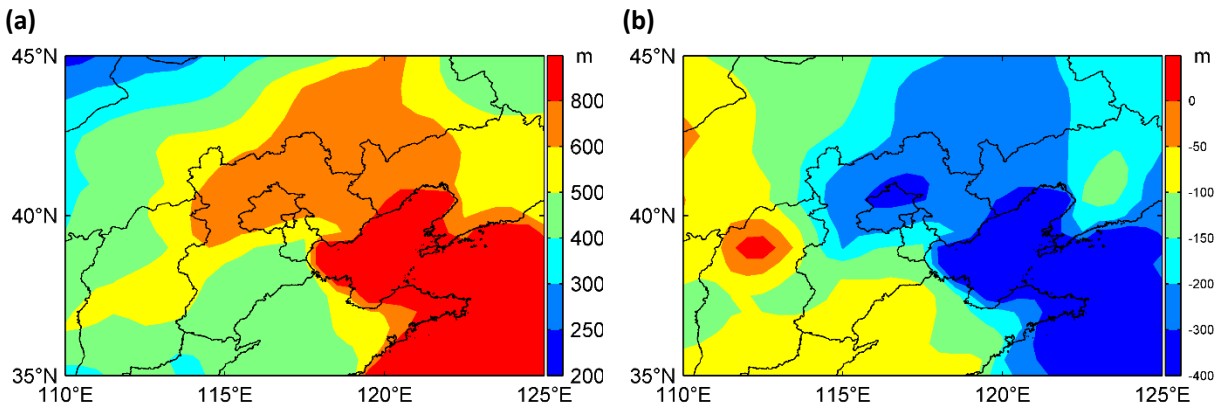

Figure 8: (a) The monthly mean BLH in December 2014 and (b) the change of monthly average
BLH in December between 2015 and 2014 over North China.

## 7  5. Modeling Analysis

In order to further explore and quantify the meteorological impact on the changes of the air
pollution situation between Decembers of 2014 and 2015, a comparison of two year's
simulations with the same emission data was performed for December. The differences of the
results in any air pollutants can be attributed to the difference impacted by the meteorological
conditions.
The CUACE is an atmospheric chemistry module including an emission modeling system,
gaseous/aerosol and chemistry processes, as well as related thermodynamic equilibrium
modules for processing the transformation between gas and particle matter (Gong et al., 2003;
Wang et al., 2010; Zhou et al., 2012). The meteorological model coupled by CUACE is the fifth-
generation Penn State/NCAR mesoscale model (MM5). The model system MM5/CUACE was run
with three nested domains (a horizontal resolution of 27 km, 9 km, and 3 km) to reduce spurious
inner domain boundary effects (Figure S1). In the vertical, there are a total of 35 full eta levels
extending to the model top at 10 hPa, with 16 levels below 2 km. One month (December) of
simulation was done for 2014 and 2015, respectively, with the result differences presented for
the analysis. The comparison of the CUACE emission inventory (representing the emission in 2013)
to other inventories, and the details of the integration scheme, initial and boundary conditions
were presented in He et al. (2016b).
Six statistical indices, i.e., index of agreement (IOA), correlation coefficient (R), standard
deviation (STD), root mean square error (RMSE), mean bias (MB), and mean error (ME) were
employed to investigate the performance of MM5/CUACE, with the routine meteorological data
from CMA, and hourly average $PM_{2.5}$ concentrations from the Ministry of Environmental
Protection. The statistical performance based on hourly observed data were provided in Table S1
and Table S2 for MM5 and CUACE, respectively. Directly comparisons between observed and
simulated daily average $PM_{2.5}$ concentrations are shown in Figure S2. The error in December 2015
is larger than that in December 2014, which maybe relate to the uncertainty of emission
inventory which represents the emission in 2013. The MB of $PM_{2.5}$ reached 25-30 $\mu g\ m^{-3}$ in the
simulation for July and December 2013 (He et al., 2016), while it decreases to 19 and 17 $\mu g\ m^{-3}$
for December 2014 and 2015 respectively (Table S2), indicating that the emission in CUACE model
might be overestimated considering the emission reduction gradually in recent years. The error
of simulated meteorological fields is another important source for the error of simulated $PM_{2.5}$
concentrations. In generally, MM5 and CUACE model can well reproduce the variation
characteristics of meteorological parameters and air pollution, and is comparable with previous
studies (He et al., 2016b; Kioutsioukis et al., 2016).

3        Figure 9a shows the December PM$_{2.5}$ concentration difference between 2015 and 2014. It is

clear that the meteorological conditions alone have contributed to the worsening air quality
(PM$_{2.5}$) in Northern China, with a high degradation of about 30-60 µg m$^{-3}$ in the southern Beijing
and southern Hebei regions in December 2015, corresponding well with WSCL from the surface
meteorological data analysis (Fig. 4), which indicates the more stable zone moving to closer to
southern Beijing.

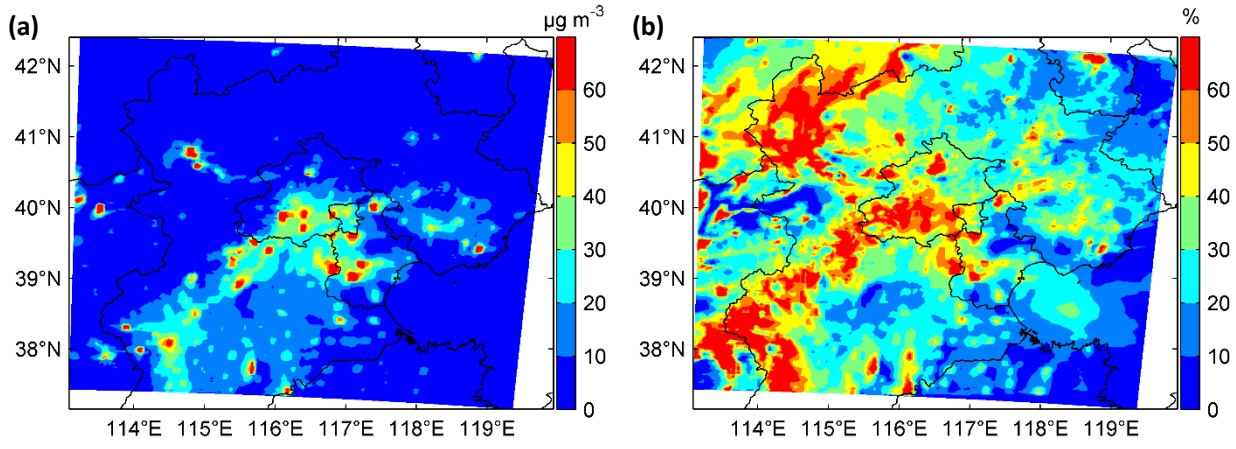

Figure 9: (a) Simulated PM$_{2.5}$ concentrations difference between December of 2015 and 2014. (b)
PM$_{2.5}$ fractional difference.

14       From the modeling results, it can also be found out that the PM$_{2.5}$ difference percentage (i.e.,

the concentration difference between December 2015 and 2014 divided by the average
concentration in December 2014) due to meteorological difference between December 2014 and
2015 for the major cities in Jing-Jin-Ji area is in the range of 10-150% (Fig. 9b), a system-wide
negative impacts on air quality in the region in 2015. This simulated difference is a comprehensive
consequence of the meteorological impacts, including the circulation, dispersing ability,
deposition, transports and chemical reactions.
It is well known that the $PM_{2.5}$ concentrations are determined by three major factors:
emissions, meteorology and atmospheric processes. Given that the degree of meteorological
impacts was simulated by the model as well as the observed differences between December 2014
and 2015 were known, the impact from emission changes can be inferred from the observed
differences and the simulated meteorological impacts.
Table 2 is a summary of the difference for major cities in Jing-Jin-Ji area between December
2014 and 2015. The observed percentage changes are all smaller than those by simulations
except Beijing, indicating that if no emission controls measures were taken during this period,
the observed difference would be much larger than the reality. Therefore, it can be deduced that
despite of the un-favorite weather conditions that worsened the air quality in December 2015,
the control measures have made a great contribution to reduce the ambient concentrations with
about 9% in Jing-Jin-Ji area. The increase of relative variation of simulated $PM_{2.5}$ concentration
for December between 2015 and 2014 in Beijing is larger than observed value, which might be
related to the bias of local wind field. In fact, it is very difficult for mesoscale meteorological
model to capture the local wind field very exactly. The comparison of wind rose map between
observation and simulation in December 2015 over Beijing (Figure S4) reveals that MM5
overestimated the frequency of northwesterly wind, which results in the underestimation of
regional transport and $PM_{2.5}$ concentration in the model. This can explain that there is a
difference of relative change of $PM_{2.5}$ in Beijing with other cities (Table 2). Another factor needs
further investigation is the observational data quality itself, which may have large uncertainty
associated from different stations and may influence the accurate assessment of the impact.
Table 2: Comparison of observed and simulated PM$_{2.5}$ in December 2015 and 2014

| City | Observed PM$_{2.5}$ (μg m-3) | | | Simulated PM$_{2.5}$ (μg m-3) | | |
|---|---|---|---|---|---|---|
| | 2014 | 2015 | Diff (%) | 2014 | 2015 | Diff (%) |
| Beijing | 61 | 145 | 137 | 68 | 114 | 68 |
| Tianjin | 113 | 125 | 10 | 92 | 129 | 40 |
| Shijiazhuang | 121 | 158 | 30 | 83 | 125 | 51 |
| Jing-Jin-Ji | 102 | 139 | 36 | 82 | 119 | 45 |

## 6. Conclusions

7       The meteorological data analysis and modeling study of 2015 winter heavy haze pollution

episodes were carried out to explore the causes of the unusual increase of haze (PM$_{2.5}$) in
December. It is found out that the monthly mean PM$_{2.5}$ concentrations in December 2015 saw a
large increase compared to the same month in 2014, ranging from 5% to 137% in Jing-Jin-Ji area,
with a mean increase of 36%. As unusual atmospheric circulation in winter 2015 (El Niño event),
the warm and wet flow has been enhanced in North China and the WSCL has shifted northerly
compared to that in 2014. The SWH and RH2 increase 12 and 25% in Jing-Jin-Ji area, respectively.
These changes of meteorology brought more static stable weather, which was the primary
responsibility for degradation of air pollution in winter 2015. Modeling analysis further confirmed
that the meteorological conditions contributed to the worsening air quality in the Jing-Jin-Ji area
in winter 2015, with the $PM_{2.5}$ concentration for the major cities in December 2015 increased 45%
compared to the same period of 2014. With the same emission data in the modeling study for
2014 and 2015, the relative changes of pollution level between two years were larger than those
from the observation, indicating the control measures have effectively brought the $PM_{2.5}$ down
(~9%) to compensate the negative meteorological impacts.
Acknowledgments
This research was financially supported by the Science and Technology support program
(2014BAC16B03) and by the National Natural Science Foundation of China (No.51305112).

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
