# Peer review of "Attributions of meteorological and emission factors to the 2015 winter severe"

_Atmospheric Chemistry and Physics, 2016_

## Referee Comment (RC1) · Anonymous Referee #1 · 16 Oct 2016

General comments:

This work aims at attributing impacts of meteorology and emissions to the formation of severe air pollution episodes in northern China between 2014 and 2015, the latter of which seeing worsened wintertime pollution regardless of improved air quality in earlier months. The manuscript is well organized and clearly presented. While it is well known that both emission and meteorology control the level of air pollution, quantitative analysis has rarely been conducted. This work presents employs ground observations and atmospheric modeling to differentiate the contribution by each factor, providing scientific insight to similar phenomena elsewhere. There are a few issues concerning the analytical approach used here and the conclusions drawn from these analyses that

need to be adequately addressed before it can be considered for publication at ACP.

Major comments:

1. The approach for meteorological factor analysis: It is an interesting approach to examine the linkage between weather pattern and PM2.5 level. It seems that some meteorological parameters are more strongly associated with PM2.5 concentrations than others. The analysis can be enhanced if these associations can be illustrated by comparing the correlations between PM2.5 and each parameter (wind speed, wind direction, temperature, and relative humidity) in the two years.

2. Considerable uncertainty may be associated with the indirect method utilized to quantify emission contribution to wintertime PM2.5 changes between the two years. The emission contribution is derived from the difference between simulated and observed PM2.5 changes under the meteorological conditions representing 2014 and 2015. The model simulations are subject to uncertainty in predicting both meteorological parameters and PM2.5 concentration (e.g., Table 1). It may be useful to address these uncertainties by first evaluating the model skills to predict key meteorological parameters.

3. A more direct method to quantify emission contribution would be to conduct additional simulations by using emission data representing the two winters. A challenge of such a method is to obtain accurate emission trends for key precursors to PM2.5. It may be possible to derive such trends from the ground or satellite observations near emission sources or during particular time window (e.g., 6-9am local time for mobile sources from near road monitors). Adding these additional analyses will make the emission attribution more convincing.

Specific comments:

Page 2: L2-3 with an annual mean concentration of fine particulate matter (PM2.5) ranging from . . .

[Figure]

L6: change "negative" to "adverse".

L12: change drop to dropping;

L18: change "hardly combined" to "did not consider";

Page 3: L11-12: study of air pollution conditions in the last two months of 2015

L15: give abbreviations for both terms first used here.

Page 6: Figure 2. It is difficult to discern the numbers and text in these maps. Please simplify the background and highlight the text/numbers relevant to the main message here.

Page 7: Table 1. If we use SO2 as an indicator to coal burning emission sources and NOx to mobile sources, it seems that either coal burning was significantly lower in 2015, or SO2 to sulfate conversion was more efficient, regardless of increased concentrations in CO and NO2 from 2014 to 2015. How important is sulfate to PM2.5 in these cities? If chemical speciated measurements of PM2.5 are available during this study, it would interesting to analyze the SO2 to sulfate ratio and NO2 to nitrate ratio to see if the gas-to-particle conversion has changed over time. It will be useful to understand the relative contribution from emissions, transport, or gas-to-particle processes.

Page 8: L2. How was WSCL calculated here? Please provide either details of the calculation or a traceable reference.

Page 9: L1. Please clarify "temperature anomaly". Is it higher or lower than the average?

L4. northward?

L14. Remove "were".

Page 10: L9: 2015

Page 11: L2: remote "was" L13: more than 27% or doubled?

Page 13: L9: emission modeling system; L14: mode(l)

P15: L1: define how was the difference percentage calculated.

P16: L1-3: the 8% seems be reasonable for emission changes from one year to the next, but the number is very small considering the large changes and opposite directionality in PM2.5 precursor concentrations.

Table 3: large discrepancy exists between observed and simulated PM2.5 concentrations. What are the major reasons underlying these biases and how will the biases be propagated into the met/emission attribution?

---

## Referee Comment (RC2) · Anonymous Referee #2 · 17 Oct 2016

This manuscript presents a comprehensive analysis of the differences in meteorological conditions over North China between winter 2014 and 2015 and the impact of those differences on PM2.5 air quality. The analysis is solid and the presentation is clear and well-written. I recommend publication in ACP after the following comments are addressed.

Major comment:

1. More evidence or discussion is needed to support the authors' argument that the meteorological difference between winter 2014 and 2015 over North China is related

to the strong El Nino in 2015. The analysis presented in the manuscript is all confined to North China, so it's difficult to judge whether the meteorological difference is due to the intrinsic year to year variability on local and regional scale or indeed something related to ENSO. For example, the authors can discuss if prior El Nino winters have seen similar changes in the wind convergence zone over North China. In addition, previous analyses have analyzed the role of winter monsoon on wintertime PM pollution in North China (e.g. Jia et al., A new indicator on the impact of large-scale circulation on wintertime particulate matter pollution over China, ACP, 2015). The authors should give a more comprehensive summary of those prior studies that link region- and local-scale meteorology changes to larger-scale variability. In this context, this manuscript might be the first to investigate the role of El Nino on winter time PM pollution in China.

2. The manuscript focuses on North China, but only four cities are analyzed. Are these cities representative of the whole region? What is the specific domain of North China? Better to include a regional mean comparison between the two winters in the tables and discussion.

Minor comments:

1. Pg 2, line 12: change 'drop' to 'dropping'

2. Pg 3, line 5: change "as high" to "as high as"

3. Figure 1: missing symbols and legends

4. Pg 7, line 6: what are the emissions control measures implemented between the two years? How significantly are these measures expected to reduce emissions? It will be helpful to describe them in the context of changing pollution levels between the two years.

5. Pg 7, line 14: change "gathering" to "accumulation"

---

## Referee Comment (RC3) · Anonymous Referee #3 · 18 Nov 2016

**Comments on "Attributions of meteorological and emission factors to the 2015 winter severe haze pollution episodes in Northern China" by T. Liu et al.**

Anonymous reviewer

18 November, 2016

**General Comments:**

This manuscript is an interesting study of the relative roles of emission and meteorology for air pollution. Through data analysis and numerical experiments, the authors conclude that unfavourable meteorological condition was the main reason for the 2015 winter severe haze episodes in Northern China, and emission control alleviated the air pollution. In general this manuscript is well organised and provides some new insights. However, some conclusions are not adequately supported by the modelling and data analysis. Many important details (e.g. model validation and uncertainty quantification) are not presented, and some method and model results need to be further discussed. There are a few major issues and concerns that should be addressed.

1) The authors conduct numerical experiments to investigate the effectiveness of meteorology versus emission. To ensure convincing results, the well-simulated meteorology and particulate matter is prerequisite. The model skill can be evaluated using multi-site measurements. The simulated meteorological parameters and particulate pollutants should be compared with observing data in temporal contrast, and statistical analysis is helpful for the validation. Besides, the uncertainty quantification of the results needs to be performed.

2) A large disparity was found between the mean observed and simulated PM2.5 concentration (in Table 3). How about the time and space variations of the simulated bias? What were the main sources of error?

3) The limitations and uncertainties in the approach of quantifying of meteorological contribution needs to be addressed. For example, one of the underlying assumptions for the approach is that amount and spatiotemporal distribution of the precursor sources emission are accurate; but it is well documented that emission inventory in Asia/China could associated with considerable uncertainty. How about the emission data used in the modelling? The potential impact of the emission uncertainty on the quantification results needs to be discussed.

4) Please re-evaluate the reliability of quantifying the contribution of emission change since the indirect method may have pronounced uncertainty.

5) Traceability is important for a scientific paper. Many important details (e.g. model configuration) should be included in the manuscript.

6) How were the wind speed convergence lines (WSCL) calculated? Could model re-produce the observed WSCL? Please provide the supporting materials for the WSCL analyses.

**Specific comments:**

1) Abstract: The authors are advised to present the method and quantitative results in brief here.

2) Line 17: "meteorology" —> "meteorological"

3) Page 3, Line 6-8: Zhao et al is not found in the reference list. If possible, please add references on the association of haze with ENSO.

4) The authors are advised to limit the number of cited non-English references.

5) P8, Line2: In my opinion, "wind convergence line" is more accurate than "wind speed convergence line" (WSCL).

6) -, 2nd paragraph: The analyses in this paragraph need sufficient supporting materials, and the conclusions need further discussed. The association of ENSO with east asian winter monsoon and shifting of the WSCL can not be simply established in the analysis.

7) P8, L19: "Research [*Si et al*., 2016]" -> "Si et al. (2016)"

8) P9, L2: How to draw the conclusion "the cold front in 2015 could not extend to the degree as in 2014"? It is not completely reasonable. The mesoscale cold fronts are embedded in the extratropical weather systems (i.e. baroclinic waves and the associated extratropical cyclones). Usually, the cold fronts exhibit fast movement and low occurrence frequency (about 5~10 days per time) in the mid-latitude region. Hence, the cold front locations are not suitable to be monthly averaged for this analysis. As far as one case was concerned, the cold front in 2015 could extend to that degree/location.

9) PBL height is an important meteorological parameter for atmospheric transport and dispersion. The meteorological factors will be more comprehensively considered if the analysis of PBL height is added.

10) P13, L13: Please briefly address the "2010 HTAP emission inventory data" and spatiotemporal variations of "hourly-gridded data".

11) -, L14-17: Many model's parameters (e.g., grid number, vertical levels and model initialization) should be clarified.

12) References: Check and correct the reference format. Non-English reference should be given clear indication of the language (for example, "in Chinese").

13) Some new papers (e.g., Wang et al., 2016; Yang et al., 2016) are closely relevant to this work. The authors are advised to cite or discuss these works.

References:

1) Wang, X., Wang, K. and Su, L.: Contribution of Atmospheric Diffusion Conditions to the Recent Improvement in Air Quality in China, Sci Reports, 6, 36404, doi:10.1038/srep36404, 2016.

2) Yang, Y., Liao, H. and Lou, S.: Increase in winter haze over eastern China in recent decades: Roles of variations in meteorological parameters and anthropogenic emissions, J Geophys Res Atmospheres, doi:10.1002/2016JD025136, 2016.

---

## Author Comment (AC1) · 23 Jan 2017

1. The approach for meteorological factor analysis: It is an interesting approach to examine the linkage between weather pattern and PM2.5 level. It seems that some meteorological parameters are more strongly associated with PM2.5 concentrations than others. The analysis can be enhanced if these associations can be illustrated by comparing the correlations between PM2.5 and each parameter (wind speed, wind direction, temperature, and relative humidity) in the two years.

Response: Thanks for the advice. The correlations between the daily average PM2.5 concentrations and daily average meteorological parameters during 2014-2015 are added in Figure 3 in the revised manuscript. PM2.5 concentrations are positively correlated with 2-m temperature and relatively humidity, while negatively correlated with 10-m wind speed and boundary layer height. The correlation coefficients are significant except correlation for 2-m temperature in Shijiazhuang.

2. Considerable uncertainty may be associated with the indirect method utilized to quantify emission contribution to wintertime PM2.5 changes between the two years. The emission contribution is derived from the difference between simulated and observed PM2.5 changes under the meteorological conditions representing 2014 and 2015. The model simulations are subject to uncertainty in predicting both meteorological parameters and PM2.5 concentration (e.g., Table 1). It may be useful to address these uncertainties by first evaluating the model skills to predict key meteorological parameters.

Response: A new set of numerical simulations were conducted with the new results used in the revised manuscript. Six statistical indices, i.e., index of agreement (IOA), correlation coefficient (R), standard deviation (STD), root mean square error (RMSE), mean bias (MB), and mean error (ME), were employed to investigate the performance of modeling system (Table S1 and Table S2). In general, the model can well reproduce the variation characteristics of meteorological parameters and air pollutant levels, which are comparable with previous studies (He et al., 2016; Kioutsioukis et al., 2016). Reference: He J.J., Wu L., Mao H.J., Liu H.L., Jing B.Y., Yu Y., Ren P.P., Feng C., Liu X.H.: Development of a vehicle emission inventory with high temporal-spatial resolution based on NRT traffic data and its impact on air quality in Beijing-Part 2: Impact of vehicle emission on urban air quality. Atmos. Chem. Phys., 16, 3171–3184, doi:10.5194/acp-16-3171-2016, 2016. Kioutsioukis, I., de Meij, A., Jakobs, H., Katragkou, E., Vinuesa, J., and Kazantzidis, A.: High resolution WRF ensemble forecasting for irrigation: Multi-variable evaluation, Atmos. Res., 167, 156-174, doi:10.1016/j.atmosres.2015.07.015, 2016.

3. A more direct method to quantify emission contribution would be to conduct additional simulations by using emission data representing the two winters. A challenge of

such a method is to obtain accurate emission trends for key precursors to PM2.5. It may be possible to derive such trends from the ground or satellite observations near emission sources or during particular time window (e.g., 6-9am local time for mobile sources from near road monitors). Adding these additional analyses will make the emission attribution more convincing.

Response: We agree that this is another way to assess the emission contributions. However, it is almost impossible to get the accurate emissions for different chemical species from ground or satellite data. As a matter of fact, this paper is trying to use the model to get such information.

4. Page 2: L2-3 with an annual mean concentration of fine particulate matter (PM2.5) ranging from . . . Response: It has been modified according to the suggestion.

5. P2L6: change "negative" to "adverse". Response: It has been modified according to the suggestion.

6. P2L12: change drop to dropping Response: It has been modified according to the suggestion.

7. P2L18: change "hardly combined" to "did not consider" Response: It has been modified according to the suggestion.

8. Page 3: L11-12: study of air pollution conditions in the last two months of 2015 Response: It has been modified according to the suggestion.

9. P3L15: give abbreviations for both terms first used here. Response: The abbreviations appear in the revised manuscript has been checked carefully.

10. Page 6: Figure 2. It is difficult to discern the numbers and text in these maps. Please simplify the background and highlight the text/numbers relevant to the main message here. Response: The Figure has been replotted in the revised manuscript.

11. Page 7: Table 1. If we use SO2 as an indicator to coal burning emission sources

and NOx to mobile sources, it seems that either coal burning was significantly lower in 2015, or SO2 to sulfate conversion was more efficient, regardless of increased concentrations in CO and NO2 from 2014 to 2015. How important is sulfate to PM2.5 in these cities? If chemical speciated measurements of PM2.5 are available during this study, it would interesting to analyze the SO2 to sulfate ratio and NO2 to nitrate ratio to see if the gas-to-particle conversion has changed over time. It will be useful to understand the relative contribution from emissions, transport, or gas-to-particle processes.

Response: Thanks for your advice. The chemical species can help us to understand the relative contribution from emissions and atmospheric chemical processes. Unfortunately, there isn't chemical observed data for us in December 2014 and 2015.

12. Page 8: L2. How was WSCL calculated here? Please provide either details of the calculation or a traceable reference.

Response: Wind speed sheer, i.e., abrupt decrease (increase) of wind speed, forms a convergence (divergence) zone. Based on weather analysis method, the WSCL was identified according to wind speed sheer line. The instruction of WSCL has been provided in the revised manuscript.

13. Page 9: L1. Please clarify "temperature anomaly". Is it higher or lower than the average? Response: It has been modified to make the description clearer.

14. P9L4. northward? Response: It has been modified according to the suggestion.

15. P9L14. Remove "were". Response: It has been modified according to the suggestion.

16. Page 10: L9: 2015 Response: Thanks for your reminding. It has been corrected.

17. Page 11: L2: remote "was" L13: more than 27% or doubled? Response: It has been modified according to the suggestion.

18. Page 13: L9: emission modeling system; L14: mode(l) Response: It has been

modified according to the suggestion.

19. P15: L1: define how was the difference percentage calculated. Response: An explanation has been provided in the revised manuscript.

20. P16: L1-3: the 8% seems be reasonable for emission changes from one year to the next, but the number is very small considering the large changes and opposite directionality in PM2.5 precursor concentrations.

Response: Thanks for your advice. The values of PM2.5 concentration variation between December of 2014 and 2015 due to emission control have some uncertainty caused by uncertainty of air quality simulation. Hence these descriptions have been removed in the revised manuscript.

21. Table 3: large discrepancy exists between observed and simulated PM2.5 concentrations. What are the major reasons underlying these biases and how will the biases be propagated into the met/emission attribution?

Response: New numerical simulation was conducted, which was introduced in the revised manuscript. The comparison between simulated and observed PM2.5 concentration (Figure S2), and the statistical analysis reveal that CUACE model can well reproduce the variation characteristics of PM2.5 concentration. Large discrepancy between observed and simulated PM2.5 concentrations was caused mostly by uncertainty of emission inventory. The discussion about the performance of CUACE has been provided in the revised manuscript.

Please also note the supplement to this comment:
http://www.atmos-chem-phys-discuss.net/acp-2016-798/acp-2016-798-AC1-supplement.pdf

[Figure]

**Supplement:**

Table 1 Performance statistics of near surface meteorological parameters.

| | IOA | R | $STD_O$ | $STD_F$ | RMSE | MB | ME |
|---|---|---|---|---|---|---|---|
| $T_2{}^a$ | 0.83 | 0.88 | 3.7 K | 4.8 | 3.6 | -2.8 | 3.0 |
| $Q_2{}^a$ | 0.52 | 0.73 | 0.5 g kg$^{-1}$ | 0.6 | 1.0 | 1.0 | 1.0 |
| $WD_{10}{}^a$ | 0.66 | 0.41 | 111.6 | 88.8 | 119.8 | 47.1 | 74.8 |
| $WS_{10}{}^a$ | 0.74 | 0.62 | 1.6 m s$^{-1}$ | 1.4 | 1.5 | 0.7 | 1.2 |
| $T_2{}^b$ | 0.82 | 0.84 | 3.2 K | 4.5 | 3.4 | -2.3 | 2.8 |
| $Q_2{}^b$ | 0.76 | 0.79 | 0.7 g kg$^{-1}$ | 0.7 | 0.7 | 0.6 | 0.6 |
| $WD_{10}{}^b$ | 0.61 | 0.29 | 118.7 | 109.1 | 139.7 | 32.1 | 93.7 |
| $WS_{10}{}^b$ | 0.81 | 0.68 | 1.5 m s$^{-1}$ | 1.4 | 1.2 | 0.3 | 0.9 |

[a] and [b] represent December 2014 and 2015 respectively.

Table 2 Performance statistics of hourly $NO_2$ and $PM_{2.5}$ concentrations in December 2014 and 2015.

| | IOA | R | $STD_O$ | $STD_F$ | RMSE | MB | ME |
|---|---|---|---|---|---|---|---|
| $PM_{2.5}{}^a$ | 0.66 | 0.52 | 91.9 μg m$^{-3}$ | 76.3 μg m$^{-3}$ | 87.3 μg m$^{-3}$ | -18.8 μg m$^{-3}$ | 59.3 μg m$^{-3}$ |
| $PM_{2.5}{}^b$ | 0.66 | 0.48 | 113.0 μg m$^{-3}$ | 108.7 μg m$^{-3}$ | 112.2 μg m$^{-3}$ | -16.5 μg m$^{-3}$ | 83.2 μg m$^{-3}$ |

[a] and [b] represent December 2014 and 2015 respectively.

[Figure]

Figure S1 The three nested domains for simulation with horizontal resolutions of 27 km, 9 km and 3 km. The colour bar represents altitude, the white line represents the administrative boundaries of province.

[Figure]

Figure S2. Comparison between observed (OBS) and simulated (FCT) daily average PM2.5 concentration in Beijing (a-b), Tianjin (c-d), and Shijiazhuang (e-f) for December 2014 and 2015 respectively.

[Figure]

Figure S3 Time series of monthly averaged SSTA over the Nino3 region in December 1979-2015.

[Figure]

Figure S4 The observed and simulated wind frequency and directions averaged in December 2015.

---

## Author Comment (AC2) · 23 Jan 2017

1. More evidence or discussion is needed to support the authors' argument that the meteorological difference between winter 2014 and 2015 over North China is related to the strong El Nino in 2015. The analysis presented in the manuscript is all confined to North China, so it's difficult to judge whether the meteorological difference is due to the intrinsic year to year variability on local and regional scale or indeed something related to ENSO. For example, the authors can discuss if prior El Nino winters have seen similar changes in the wind convergence zone over North China. In addition, previous analyses have analyzed the role of winter monsoon on wintertime PM pollution in North China (e.g. Jia et al., A new indicator on the impact of large-scale circulation

on wintertime particulate matter pollution over China, ACP, 2015). The authors should give a more comprehensive summary of those prior studies that link region- and local-scale meteorology changes to larger-scale variability. In this context, this manuscript might be the first to investigate the role of El Nino on winter time PM pollution in China.

Response: Very good question. To fully answer these questions, we have used the EAR-Interim data from European Centre for Medium-Range Weather Forecasts (ECMWF) in December 1979-2015, including sea surface temperature (SST), mean sea level pressure (MSL), 2-m temperature (T2), 10-m U and V wind component (U10 and V10), to investigate the relationship between ENSO and the air quality in North China. Area averaged SST anomalies (SSTA) over the Nino3 region (5°N-5°S, 150°-90°W) provide an index typically used to represent ENSO variability (Tang et al., 2016). Time series of monthly averaged SSTA over the Nino3 region are shown in Figure S3. Significant ENSO events were found in 1982, 1997 and 2015. The MSL and 10-m wind anomalies over North China region are shown in Figure 7 in the revised manuscript. It is found that ENSO (SSTA>0) results in weaker cold air and northerly wind, vice versa for the La Nina (SSTA<0) periods. These relationships indicate that the worse air quality in December 2015 over North China was correlated with significant ENSO event. Reference: Tang Y. L., Li L. J., Dong W. J., and Wang, B.: Tracing the source of ENSO simulation differences to the atmospheric component of two CGCMs. Atmospheric Science Letters, doi: 10.1002/asl.637, 2016.

2. The manuscript focuses on North China, but only four cities are analyzed. Are these cities representative of the whole region? What is the specific domain of North China? Better to include a regional mean comparison between the two winters in the tables and discussion.

Response: To make study area clearer, the revised manuscript focuses on the severe haze pollution in December 2015 over Jing-jin-ji area. Three important cities, i.e., Beijing, Tianjin, and Shijiazhuang, were selected for the analysis. The regional mean comparison over 13 cities has also been supplied in the revised manuscript.

3. Pg 2, line 12: change 'drop' to 'dropping' Response: It has been corrected in the manuscript.

4. Pg 3, line 5: change "as high" to "as high as" Response: It has been corrected in the manuscript.

5. Figure 1: missing symbols and legends Response: It has been corrected in the revised manuscript.

6. Pg 7, line 6: what are the emissions control measures implemented between the two years? How significantly are these measures expected to reduce emissions? It will be helpful to describe them in the context of changing pollution levels between the two years.

Response: Thanks for your suggestion. The emission control measures implemented in Jingjinji area have been provided in the revised manuscript. Although the emission reduction in 2015 over China has been released to the public (http://www.zhb.gov.cn/hjzl/zghjzkgb/lnzghjzkgb/), the emission reduction over Jingjinji area is still unknown.

7. Pg 7, line 14: change "gathering" to "accumulation" Response: It has been modified in the manuscript.

Please also note the supplement to this comment:
http://www.atmos-chem-phys-discuss.net/acp-2016-798/acp-2016-798-AC2-supplement.pdf

**Supplement:**

Table 1 Performance statistics of near surface meteorological parameters.

| | IOA | R | $STD_O$ | $STD_F$ | RMSE | MB | ME |
|---|---|---|---|---|---|---|---|
| $T_2{}^a$ | 0.83 | 0.88 | 3.7 K | 4.8 | 3.6 | -2.8 | 3.0 |
| $Q_2{}^a$ | 0.52 | 0.73 | 0.5 g kg$^{-1}$ | 0.6 | 1.0 | 1.0 | 1.0 |
| $WD_{10}{}^a$ | 0.66 | 0.41 | 111.6 | 88.8 | 119.8 | 47.1 | 74.8 |
| $WS_{10}{}^a$ | 0.74 | 0.62 | 1.6 m s$^{-1}$ | 1.4 | 1.5 | 0.7 | 1.2 |
| $T_2{}^b$ | 0.82 | 0.84 | 3.2 K | 4.5 | 3.4 | -2.3 | 2.8 |
| $Q_2{}^b$ | 0.76 | 0.79 | 0.7 g kg$^{-1}$ | 0.7 | 0.7 | 0.6 | 0.6 |
| $WD_{10}{}^b$ | 0.61 | 0.29 | 118.7 | 109.1 | 139.7 | 32.1 | 93.7 |
| $WS_{10}{}^b$ | 0.81 | 0.68 | 1.5 m s$^{-1}$ | 1.4 | 1.2 | 0.3 | 0.9 |

[a] and [b] represent December 2014 and 2015 respectively.

Table 2 Performance statistics of hourly $NO_2$ and $PM_{2.5}$ concentrations in December 2014 and 2015.

| | IOA | R | $STD_O$ | $STD_F$ | RMSE | MB | ME |
|---|---|---|---|---|---|---|---|
| $PM_{2.5}{}^a$ | 0.66 | 0.52 | 91.9 μg m$^{-3}$ | 76.3 μg m$^{-3}$ | 87.3 μg m$^{-3}$ | -18.8 μg m$^{-3}$ | 59.3 μg m$^{-3}$ |
| $PM_{2.5}{}^b$ | 0.66 | 0.48 | 113.0 μg m$^{-3}$ | 108.7 μg m$^{-3}$ | 112.2 μg m$^{-3}$ | -16.5 μg m$^{-3}$ | 83.2 μg m$^{-3}$ |

[a] and [b] represent December 2014 and 2015 respectively.

[Figure]

Figure S1 The three nested domains for simulation with horizontal resolutions of 27 km, 9 km and 3 km. The colour bar represents altitude, the white line represents the administrative boundaries of province.

[Figure]

Figure S2. Comparison between observed (OBS) and simulated (FCT) daily average PM2.5 concentration in Beijing (a-b), Tianjin (c-d), and Shijiazhuang (e-f) for December 2014 and 2015 respectively.

[Figure]

Figure S3 Time series of monthly averaged SSTA over the Nino3 region in December 1979-2015.

[Figure]

Figure S4 The observed and simulated wind frequency and directions averaged in December 2015.

---

## Author Comment (AC3) · 23 Jan 2017

1. The authors conduct numerical experiments to investigate the effectiveness of meteorology versus emission. To ensure convincing results, the well-simulated meteorology and particulate matter is prerequisite. The model skill can be evaluated using multi-site measurements. The simulated meteorological parameters and particulate pollutants should be compared with observing data in temporal contrast, and statistical analysis is helpful for the validation. Besides, the uncertainty quantification of the results needs to be performed.

Response: Six statistical indices, i.e., index of agreement (IOA), correlation coefficient (R), standard deviation (STD), root mean square error (RMSE), mean bias (MB), and

mean error (ME), were employed to investigate the performance of the modeling system (Table S1 and Table S2). Direct comparison between observed and simulated daily average PM2.5 concentrations is shown in Figure S2. In general, the model can well reproduce the variation characteristics of meteorological parameters and air pollution, and are comparable with previous studies. See the response to Q2 for reviewer 1.

2. A large disparity was found between the mean observed and simulated PM2.5 concentration (in Table 3). How about the time and space variations of the simulated bias? What were the main sources of error?

Response: A new set of numerical simulations were conducted, which was introduced in the revised manuscript. The comparison between simulated and observed PM2.5 concentration (Figure S2), and the statistical analysis reveal that the model can well reproduce the variation characteristics of PM2.5 concentration. The emission inventory used in the mdoel represents the emission in 2013. It is very difficult to acquire the near real time pollutant emission. The error of simulated PM2.5 concentrations is partially caused by the uncertainty of emission inventory. The error of simulated meteorological fields is another important source for the error of simulated PM2.5 concentrations. However, the error is acceptable because it is comparable with previous studies (See the response to Q2 for reviewer 1). The content in Table 3 has been modified according to the new simulation. The bias of PM2.5 concentration from new simulation is significant less than that from old simulation, indicating relative less uncertainty for modeling analysis (section 5).

3. The limitations and uncertainties in the approach of quantifying of meteorological contribution needs to be addressed. For example, one of the underlying assumptions for the approach is that amount and spatiotemporal distribution of the precursor sources emission are accurate; but it is well documented that emission inventory in Asia/China could associated with considerable uncertainty. How about the emission data used in the modelling? The potential impact of the emission uncertainty on the quantification results needs to be discussed.

Response: The model has an independent pollution emission module, which contains natural and anthropogenic emissions including many gas and particle matter emissions (Gong et al., 2009). Anthropogenic emissions of $SO_2$, $NO_x$, CO, VOCs, $PM_{2.5}$, $PM_{10}$, BC, OC, etc. used in emission module were developed by China Meteorological Administration based on Multi-resolution Emission Inventory for China (MEIC), INTEX-B inventory, the emissions database for global atmospheric research (EDGAR) and environmental statistics database. Some old data was corrected or updated according to the variation rate of anthropogenic emissions from environmental statistics database. An improved emission with high temporal–spatial resolution vehicle emission over Beijing was used to replace the old vehicle emission (He et al., 2016). The emission inventory in the model simulation represents the emission in 2013. The comparison of the emission inventory (representing the emission in 2013) to other inventories was presented in He et al. (2016). More discussions about the potential impact of the emission uncertainty have been provided in the revised manuscript. On the other hands, even though the spatial distribution of emission intensity may have an impact on the pollution levels in a city, the meteorological contribution for a city with a constant emission obtained from the current approach is reasonable and is best we can do at the moment. Reference: Gong, S. L., Zhang, X. Y., Zhou, C. H., Liu, H. L., An, X. Q., Niu, T., Xue, M., Cao, G. L., and Cheng, Y. L.: Chemical weather forecasting system CUACE and application in China's regional haze forecasting, in: Proceeding of the 26th Annual Meeting of Chinese Meteorological Society, Hangzhou, 2009 . He, J., L. Wu, H. Mao, H. Liu, B. Jing, Y. Yu, P. Ren, C. Feng, and X. Liu (2016), Development of a vehicle emission inventory with high temporal–spatial resolution based on NRT traffic data and its impact on air pollution in Beijing – Part 2: Impact of vehicle emission on urban air quality. Atmos. Chem. Phys., 16, 3171–3184.

4. Please re-evaluate the reliability of quantifying the contribution of emission change since the indirect method may have pronounced uncertainty.

Response: This is a good question. We agree that there exist some uncertainties in

doing such assessment. However, the method proposed in this paper is a step forward to quantify the contribution of emission changes. Further study is needed to improve the methodology.

5. How were the wind speed convergence lines (WSCL) calculated? Could model reproduce the observed WSCL? Please provide the supporting materials for the WSCL analyses.

Response: Wind speed sheer, i.e., abrupt decrease (increase) of wind speed, forms a convergence (divergence) zone. Based on weather analysis method, the WSCL was identified according to wind speed sheer line. The instruction of WSCL has been provided in the revised manuscript.

6. Abstract: The authors are advised to present the method and quantitative results in brief here. Response: It has been modified according to the suggestion.

7. Line 17: "meteorology" —> "meteorological" Response: It has been corrected in the revised manuscript.

8. Page 3, Line 6-8: Zhao et al is not found in the reference list. If possible, please add references on the association of haze with ENSO. Response: It has been corrected. A new reference has been added in the revised manuscript.

9. The authors are advised to limit the number of cited non-English references. Response: Thanks for your suggestion. Non-English references have been marked and limited as much as possible in the revised manuscript.

10. P8,Line2: In my opinion, "wind convergence line" is more accurate than "wind speed convergence line" (WSCL).

Response: Wind sheer includes wind speed sheer and wind direction sheer. The convergence in North China is caused by wind speed sheer. So we use "WSCL" in the manuscript.

11. 2nd paragraph: The analyses in this paragraph need sufficient supporting materials, and the conclusions need further discussed. The association of ENSO with East Asian winter monsoon and shifting of the WSCL can not be simply established in the analysis.

Response: The relation between atmospheric circulation in North China and Area averaged SST anomalies (SSTA) over the Nino3 are analyzed in the revised manuscript. It seems that ENSO (SSTA>0) results in weak cold air and northerly wind, while opposite for La Nina (SSTA<0). These relations indicate that the worse air quality in December 2015 over North China maybe relate to significant ENSO.

12. P8, L19: "Research [Si et al., 2016]" -> "Si et al. (2016)" Response: It has been corrected.

13. P9, L2: How to draw the conclusion "the cold front in 2015 could not extend to the degree as in 2014"? It is not completely reasonable. The mesoscale cold fronts are embedded in the extratropical weather systems (i.e. baroclinic waves and the associated extratropical cyclones). Usually, the cold fronts exhibit fast movement and low occurrence frequency (about 5~10 days per time) in the mid-latitude region. Hence, the cold front locations are not suitable to be monthly averaged for this analysis. As far as one case was concerned, the cold front in 2015 could extend to that degree/location.

Response: We agree that for each individual cold front, the degree or location reached may vary. This paper is to discuss the averaged pollution levels, therefore, we use the averaged cold front location and degree to illustrate its impact on the area and degree of air pollutant levels in these cities between 2014 and 2015.

14. PBL height is an important meteorological parameter for atmospheric transport and dispersion. The meteorological factors will be more comprehensively considered if the analysis of PBL height is added.

Response: The correlation between PM2.5 concentrations and PBL height, and the

comparison of PBL height between December 2014 and December 2015 have been provided in the revised manuscript.

15. P13, L13: Please briefly address the "2010 HTAP emission inventory data" and spatiotemporal variations of "hourly-gridded data".

Response: To get more accurate air quality simulation, new numerical simulation was conducted in the revised manuscript. Anthropogenic emissions of SO2, NOx, CO, VOCs, PM2.5, PM10, BC, OC, etc. used in emission module were developed by China Meteorological Administration based on Multi-resolution Emission Inventory for China (MEIC), INTEX-B inventory, the emissions database for global atmospheric research (EDGAR) and environmental statistics database. Some old data was corrected or up-dated according to the variation rate of anthropogenic emissions from environmental statistics database. An improved emission with high temporal–spatial resolution vehicle emission over Beijing was used to replace the old vehicle emission. The model's parameters and emission inventory are the same as previous study (He et al., 2016).

16. L14-17: Many model's parameters (e.g., grid number, vertical levels and model initialization) should be clarified. Response: It has been provided in the revised manuscript.

17. References: Check and correct the reference format. Non-English reference should be given clear indication of the language (for example, "in Chinese"). Response: It has been corrected.

18. Some new papers (e.g., Wang et al., 2016; Yang et al., 2016) are closely relevant to this work. The authors are advised to cite or discuss these works. Response: Some views from new papers has been added in the revised manuscript.

Please also note the supplement to this comment:
http://www.atmos-chem-phys-discuss.net/acp-2016-798/acp-2016-798-AC3-supplement.pdf